# Real-Time Mobile Robot Obstacles Detection and Avoidance Through EEG Signals [note 1]

**DOI:** 10.3390/brainsci15040359

**Published:** 2025-03-30

**Authors:** Karameldeen Omer, Francesco Ferracuti, Alessandro Freddi, Sabrina Iarlori, Francesco Vella, Andrea Monteriù

**Affiliations:** 1Department of Information Engineering, Università Politecnica delle Marche, 60131 Ancona, Italy; f.ferracuti@univpm.it (F.F.); a.freddi@univpm.it (A.F.); s.iarlori@univpm.it (S.I.); f.vella@pm.univpm.it (F.V.); a.monteriu@univpm.it (A.M.); 2Mechanical Department, University of Khartoum, Khartoum 11115, Sudan

**Keywords:** brain–computer interface (BCI), EEG-based human–robot interaction, real-time obstacle detection, error-related potentials (ErrPs), steady-state visually evoked potentials (SSVEPs), assistive robotic technologies, human feedback in robotics, cognitive load in BCI systems, active and passive BCI

## Abstract

Background/Objectives: The study explores the integration of human feedback into the control loop of mobile robots for real-time obstacle detection and avoidance using EEG brain–computer interface (BCI) methods. The goal is to assess the possible paradigms applicable to the most current navigation system to enhance safety and interaction between humans and robots. Methods: The research explores passive and active brain–computer interface (BCI) technologies to enhance a wheelchair-mobile robot’s navigation. In the passive approach, error-related potentials (ErrPs), neural signals triggered when users comment or perceive errors, enable automatic correction of the robot navigation mistakes without direct input or command from the user. In contrast, the active approach leverages steady-state visually evoked potentials (SSVEPs), where users focus on flickering stimuli to control the robot’s movements directly. This study evaluates both paradigms to determine the most effective method for integrating human feedback into assistive robotic navigation. This study involves experimental setups where participants control a robot through a simulated environment, and their brain signals are recorded and analyzed to measure the system’s responsiveness and the user’s mental workload. Results: The results show that a passive BCI requires lower mental effort but suffers from lower engagement, with a classification accuracy of 72.9%, whereas an active BCI demands more cognitive effort but achieves 84.9% accuracy. Despite this, task achievement accuracy is higher in the passive method (e.g., 71% vs. 43% for subject S2) as a single correct ErrP classification enables autonomous obstacle avoidance, whereas SSVEP requires multiple accurate commands. Conclusions: This research highlights the trade-offs between accuracy, mental load, and engagement in BCI-based robot control. The findings support the development of more intuitive assistive robotics, particularly for disabled and elderly users.

## 1. Introduction

The integration of brain–computer interfaces (BCIs) into robotics or brain–machine interfaces (BMIs) has dramatically advanced human–robot interaction (HRI), facilitating more nuanced and effective communication between humans and machines. BMIs enable the direct translation of neural information into commands that robots can execute, enhancing their capabilities across various fields, including industrial automation and assistive technologies [1]. This technology plays a critical role in improving the lives of individuals with disabilities by providing advanced prosthetic control and enabling assistive robots to perform tasks that users cannot manage independently [2,3,4].

### 1.1. Overview

In the field of assistive robotics, where safety is paramount, the close and frequent interaction between the robot and the user underscores the importance of incorporating human feedback into robotic systems. This feedback is critical in collaborative tasks, where it helps mitigate factors that can impair performance and enhance safety in environments where robots operate near humans, such as homes, hospitals, and workplaces.

Physically assistive robots, including advanced smart wheelchairs and innovative walkers, rely heavily on a navigational suite that uses mapped, real-time sensor data to safely guide users to their destinations, avoiding unexpected environmental barriers or “negative obstacles” like potholes or underground areas. In this context, BCI technology has emerged as a powerful tool for integrating human intent into robotic control, allowing users to supervise various tasks through direct neural feedback [1,2,3,4,5].

EEG-based BCI has been widely explored for robot control [6,7], particularly in mobile robotics under the shared control paradigm [8]. Often, the user directly commands the robot, as demonstrated in [9], where a smart wheelchair moves forward or turns based on the user’s imagined movements. Additionally, Active BCI has been investigated to enhance autonomous navigation, enabling the system to override user commands when necessary, for instance, to steer around an obstacle safely.

To minimize user effort, BCI-controlled mobile robots can integrate autonomous navigation features. This allows for more efficient operation, freeing up users from constantly guiding the robot. In some approaches, such as [10], users guide the robot through the BCI, while autonomous functions take over during obstacle avoidance and corridor following. Despite these advancements, few studies have specifically explored enhancing a mobile robot’s obstacle avoidance capabilities through BCI methods [11], highlighting a crucial area for further research in assistive robotics.

BCIs include multiple paradigms, each with distinct applications and challenges that have significantly contributed to the advancement of HRI [12]. Motor imagery (MI) BCI allows users to control robotic limbs by imagining movements [13,14]. While it provides precise control, it demands considerable mental effort, potentially leading to fatigue [15]. P300-based BCIs rely on detecting a neural response triggered by rare or significant stimuli [16], making them effective for selecting commands in applications such as wheelchair control [17]. Though requiring minimal training, their effectiveness depends on sustained attention. Steady-state visually evoked potentials (SSVEPs) utilize visual stimuli flickering at distinct frequencies, allowing for rapid and direct control of robotic systems [18]. While SSVEPs offer high information transfer rates with minimal training requirements [19,20], they can lead to visual fatigue and are sensitive to lighting conditions. Similarly, Visually evoked potentials (VEPs) are induced by visual stimuli changes [21], offering an alternative control mechanism that still requires continuous visual attention. Lastly, Error-related potentials (ErrPs) are generated when users detect an error [22], making them valuable for passive robot correction, particularly in automation-dependent environments such as manufacturing [23]. However, ErrP detection reliability may decline with user fatigue and task complexity [24].

Recent advancements in brain–machine interfaces have introduced novel paradigms that enhance robot control and HRI, particularly through ErrPs and SSVEPs. ErrPs enable passive robot correction, allowing for autonomous error adjustments without direct user input, a crucial feature for dynamic environments requiring rapid adaptation [25]. Conversely, SSVEPs facilitate active user control by translating attention to flickering stimuli into direct commands. Despite its efficiency, prolonged SSVEP use can cause visual fatigue [25], leading to the development of improved paradigms, such as contrast-reduced stimuli, which reduce strain while maintaining efficiency [26]. Additionally, innovations like the dual frequency aggregated steady-state motion visually evoked potential (DF-SSmVEP) integrate multiple motion frequencies within a single stimulus, enhancing information transfer without increasing trial length [27]. These cutting-edge paradigms improve BMI applications in robotics, making them more efficient and user-friendly.

Beyond ErrPs and SSVEP, other BCI paradigms are employed in diverse applications. Motor imagery (MI) remains a vital method for prosthetic control, necessitating users to imagine bodily movements to manipulate robotic limbs. However, this method demands a significant amount of mental effort. P300 is widely used in applications like speller interfaces and wheelchair navigation, enabling users to select options by focusing on specific stimuli [28]. These paradigms, alongside newer approaches, contribute to refining BCI-driven assistive robotics [3].

### 1.2. Research Objectives

This study investigates how EEG-based BCI can detect and correct navigation errors in mobile robots, particularly in dynamic environments like homes, hospitals, and workplaces, where safety is critical for assistive technologies such as wheelchairs. Two key paradigms, namely passive ErrPs and active SSVEPs, are evaluated for their classification accuracy and cognitive load to determine their effectiveness in integrating human feedback into robotic navigation. Passive BCIs monitor involuntary neural responses, such as ErrPs, when users recognize a mismatch between intended and actual actions. These signals allow robots to autonomously self-correct without explicit user commands, improving adaptability in complex environments [29]. In contrast, active BCIs require users to generate brain activity for control consciously. SSVEPs exemplify this category, as users focus on flickering stimuli to issue commands. While SSVEPs offer fast, accurate control with minimal training, prolonged exposure may lead to fatigue, requiring adjustments for user comfort [30].

By identifying the most effective paradigm, this research aims to improve the design of intuitive and reliable HRI systems for assistive robotics, ensuring both functionality and user well-being. This study evaluates these paradigms not only in terms of classification accuracy for robotic navigation but also their cognitive demands, ultimately enhancing usability in real-world applications.

## 2. Materials and Methods

To achieve the research goal, we designed an experiment utilizing a simulated environment for a smart wheelchair robot. This setup incorporates human-in-the-loop approaches, enabling users to influence navigation directly. Additionally, we detailed the BCI protocols for the ErrP and SSVEP paradigms, outlining their implementation and the methods used for signal and data collection.

Finally, these BCIs are evaluated based on accuracy in both individual event detection and task completion, as well as user mental load, to identify the most effective method for integrating human feedback into the wheelchair’s navigation system, aiming to improve human–robot interaction in assistive technologies.

### 2.1. Experimental Setup and Simulation Framework

#### 2.1.1. Robot Description and Human-in-the-Loop Control

This study employs a wheelchair-mobile robot equipped with a comprehensive array of sensors, including LIDAR, cameras, an inertial measurement unit (IMU), and encoders [5,31]. These sensors enable the robot to autonomously navigate a simulated indoor environment that mimics real-world conditions, featuring a mix of static and dynamic obstacles. The simulation is run under Gazebo [32], a robust 3D simulator that allows for detailed modeling of robot interactions within a dynamically changing environment. This setup helps in testing the robot’s navigation capabilities and the effectiveness of the sensors in detecting a variety of obstacles, except those that are typically challenging like slippery surfaces, small objects, and negative obstacles such as holes or unexpected depressions in the path [33,34].

The robot’s autonomous navigation is tested in a meticulously reconstructed 3D environment of the Information Engineering Department corridor at Università Politecnica delle Marche. This environment includes artificially created holes and obstacles to assess the robot’s detection and navigation responses. The robotic control and navigation algorithms are managed through ROS (Robot Operating System), which integrates the sensor data with the robot’s motion controls.

#### 2.1.2. Human-in-the-Loop Approaches

This study integrates human feedback directly into the robot’s control loop to enhance obstacle detection and navigation. This human-in-the-loop approach is implemented in two distinct modes:

Indirect control: In this setup, the human participant acts as an auxiliary sensor that complements the robot’s onboard sensors. When the participant detects an obstacle that the robot’s sensors have missed, he/she triggers a signal. This signal initiates a process where the system estimates the obstacle’s location relative to the robot and integrates this information into the robot’s path-planning algorithms. The user interaction is simplified to pressing a key or issuing a voice command, making it accessible and straightforward [5].

Direct control: This mode offers a hands-on approach, where the participant not only detects but also navigates around the obstacle. Using a joystick, the user manually controls the robot’s movements to avoid detected hazards. This direct interaction continues until the obstacle is cleared, after which autonomous navigation resumes. This method fosters a shared control environment, blending automated navigation with manual interventions to ensure safety and precision [35].

These two modes can be summarized in Figure 1.

#### 2.1.3. Simulation and Data Integration

The experiments leverage the capabilities of Gazebo to simulate realistic navigation scenarios and the ROS (Robot Operating System) framework to manage data integration and real-time control adjustments based on human inputs. The setup allows for detailed data collection on robot performance and human interaction patterns, which are crucial for evaluating the effectiveness of the proposed human-in-the-loop approaches.

### 2.2. BCI System and Protocol

#### 2.2.1. Protocol Description and Experimental Setup

The protocol we have designed is intended to elicit EEG potentials for real-time feedback to enhance robotic navigation. These potentials are evoked when the user, observing errors during task execution, reacts cognitively to unexpected errors made by themselves or another agent. The primary focus of the protocol is to evoke the error-related potential, an event-related potential that occurs as a cognitive response to an unexpected, randomly presented sensory stimulus, capturing the user’s attention. This signal, predominantly detected in the central region of the scalp, varies in latency between 250 and 500 ms depending on the complexity of the cognitive task performed by the individual. The ErrP typically consists of two main components as detailed in the literature [36]: the first is the error-related negativity (ERN or Ne), a negative potential peaking 50–200 ms after an erroneous response; the second component, known as error positivity (Pe), is a positive potential that may occur following the ERN, with a latency of 200–500 ms, depending on the task executed.

The proposed protocol leverages the “oddball paradigm” [37], where subjects are exposed to a sequence of events classified into two categories: target stimuli and more frequent non-target events. Specifically, the protocol involves 200 trials, where short videos display a robot navigating indoor environments with obstacles. In 80% of these trials, the robot either navigates smoothly or successfully avoids obstacles by turning left or right. The remaining 20% of trials depict the robot colliding with an obstacle, such as a small hole in the floor. During the trials, participants press a specific key if they predict the robot will successfully navigate past an obstacle. EEG signals are recorded synchronously with the key presses to capture their responses. The 200 trials are divided into four groups of 50, with breaks in between to reduce fatigue.

Each group used a first-person view (FPV), which is a method of controlling a robotic or remote-controlled system where the user experiences the environment from the perspective of the machine itself. This is typically achieved by streaming videos simulating autonomous navigation from a camera mounted on the wheelchair-mobile robot, oriented toward a screen two meters ahead. Videos are presented in random order and trimmed to about four seconds using the MATLAB R2024b computer vision toolbox to vary the robot’s starting position relative to the obstacles. Each trial begins with an on-screen arrow indicating the direction the robot should turn to avoid a collision. In error trials, the robot does not follow the indicated direction and collides with the obstacle.

In an SSVEP, the used scenario is identical to that given by OpenViBE, where participants are exposed to visual stimuli flickering at specific frequencies to induce brain responses detectable in EEG signals. The experiment is configured with a display refresh rate of 60 Hz, and stimuli are presented in contrasting colors, such as red and black. Each stimulus flickers for 4 s, followed by a 2 s break to reduce fatigue. Each subject participated in two experiments totaling 40 min: 20 min of passive control using ErrPs and 20 min of active control through SSVEPs. In the first 20 min, participants focused on a passive BCI to elicit ErrPs, a type of event-related potential. After the break, subjects engaged in another 200 trials, this time using active BCI control through SSVEPs. All trials simulated autonomous robot navigation from an FPV, with the robot’s orientation being determined by a camera mounted on the wheelchair robot. The trials were randomly distributed and processed using MATLAB to ensure varied starting positions and diverse obstacle locations.

#### 2.2.2. Passive BCI (ErrP)

Participants watched the robot navigate through a simulated environment and instinctively responded to any navigational errors the robot makes, as illustrated in Figure 2. The experimental protocol includes scenarios in which the robot either successfully navigates around obstacles or fails to do so, prompting the generation of EEG signals that are synchronized with the robot’s navigation data to record the timing of observed errors accurately. These instances are captured through the “KeyDown” channel, which records key presses in real time, marking the precise latency of the events. At the end of each trial, EEG signal sequences are labeled based on consecutive trials: “Not” for successful avoidance, and “Pass” for a collision. These labels are then used for subsequent signal epoching and classification.

#### 2.2.3. Active BCI (SSVEP)

In this setup, participants actively control the robot’s movements as it approaches obstacles not detected by the sensor set by focusing on specific visual stimuli linked to various navigational commands, like turning left or moving forward, as shown in Figure 3. The visual stimuli are displayed as four squares on a screen, each flickering at distinct frequencies: 12 Hz, 15 Hz, and 20 Hz. The participants’ gaze and concentration on these flickering frequencies directly control the robot’s actions within the simulation. Additionally, a non-flickering square is included to signal periods of no intended stimulation, serving as a baseline or rest state for comparison.

#### 2.2.4. Data Acquisition and Pre-Processing

The EEG data have been recorded from 10 healthy subjects whose mean age is 28 years (standard deviation (STD) = ±3 years). The acquisition was accomplished in two different sessions: the first session was for collecting data for passive control and correcting the robot using ErrPs, for each subject recorded 200 trials, with long breaks every 50 trials; the second session was dedicated to collecting data using SSVEP protocols.

In the first session, EEG signals were recorded using BCI2000 [38] with a g.Tec g.MOBIlab+ eight-channel amplifier, along with an Acticap system. In the second session, EEG data were collected using OpenViBE software V3.5.0 [39].

A total of eight electrodes were placed according to the international 10/20 system at positions Fz, Cz, CPz, Oz, PO7, O1, O2, and Iz, as shown in Figure 4 and Figure 5. This montage effectively covered both the frontal and occipital lobes for SSVEPs, ensuring the capture of relevant neural activity. The signals were acquired at a sampling rate of 256 Hz, with a band-pass filter set between 0.5 and 30 Hz to enhance signal clarity and relevance.

Our initial approach was influenced by hardware constraints, as the g.Tec g.MOBIlab+ amplifier limited us to eight channels. However, for real-world applications, reducing the number of channels is beneficial for improving system usability, lowering costs, and minimizing computational complexity. Moving forward, we aim to refine channel selection, identifying a minimal yet effective subset that balances classification performance with practicality for real-world use.

The EEG signals are recorded together with the keyboard press (“KeyDown” channel) for event locking. The keyboard signal is binary: it is “1” when a key is pressed and “0” otherwise. This channel is used as a stimulus channel to extract all events and synchronize them together with the recorded EEG potentials. The events are then named according to the real order of the trial sequence, where the “KeyDown” channel contains the real latency only. The event names are: “Turn” for successful obstacle avoidance, “Not’’ when there are no obstacles, and “Pass’’ when the robot collides with the obstacle. These three event groups have been used for signal epoching during offline processing and classification.

#### 2.2.5. Data Processing

Data processing in this study has been performed mainly in two stages: event classification and mental load calculations. The data were then processed in two platforms: MATLAB and OpenViBe.

MATLAB is a high-level programming and numeric computing environment widely used in scientific and engineering research [40]. It offers robust capabilities for algorithm development, data visualization, data analysis, and numerical computation. Serving as the core platform for the EEGLAB [41] and BCILAB toolboxes [42], MATLAB facilitates sophisticated analyses and visualizations of EEG data and other datasets. Its comprehensive library of built-in functions and toolboxes makes it an indispensable tool across various fields, including neuroscience and robotics.

OpenViBE is an open-source software platform specifically designed for the development, testing, and application of brain–computer interfaces and real-time neurofeedback systems [39]. Initially developed by the French National Institute for Research in Digital Science and Technology (Inria) along with other collaborators, OpenViBE excels in the real-time processing of brain signals. This capability makes it highly versatile and suitable for various applications ranging from medical research to interactive gaming and immersive virtual reality environments.

A linear discriminant analysis (LDA) was selected for the event classification due to its lower computational demands and to facilitate comparison between methods, as the team has explored several classification techniques in previous work [43].

For classifying the ErrP, a two-phase temporal filtering process was implemented. Initially, EEG data underwent a second-order Butterworth low-pass filter, followed by decimation to reduce the sampling rate from 256 Hz to 32 Hz. This was succeeded by a second-order high-pass filter, resulting in a signal with a bandwidth of 3.5 to 8.5 Hz, which is ideal for capturing ErrP activities. The data were then segmented into epochs spanning from −1 s to 1 s around the event, marked by the key press. A Bayesian linear discriminant analysis (BLDA)-based classifier was subsequently employed to differentiate the events [44,45,46].

For the SSVEP classification, the common spatial pattern (CSP) algorithm was used [47]. This algorithm applied spatial filters to enhance the variance of signals from one class while suppressing that of the other, effectively simplifying the multi-channel EEG data into a manageable two-channel format. For each channel, data variance was computed and used as a feature for classification. In the SSVEP experiments, specific second-order Butterworth band-pass filters were tailored for each stimulus frequency, with cut-off values being adjusted 0.25 Hz below and above the respective stimulus frequency. The processed feature matrices were then classified using LDA to identify distinct data classes.

#### 2.2.6. Mental Load Index

In order to assess fatigue, an independent component analysis (ICA) was applied using the runica function from the EEGLAB toolbox [41]. To ensure full-ranked data, the PCA option in runica was used to match the data rank, as it has been recommended to compute the average reference after the ICA [48]. To further signal cleaning, a band-pass filter from 0.5 to 20 Hz was applied, preserving all relevant frequency bands of interest, as presented in Table 1, while excluding most muscle activity, which predominantly occurs in the 10–500 Hz range. The cleaned EEG signals were then segmented into 5 s epochs, synchronized with event triggers as zero seconds.

The analysis focused on the EEG channels Fz, Cz, and Oz due to their critical role in cognitive processing. Fz and Cz are particularly important for error detection [36], while Oz is essential for processing visual stimuli [49]. This selection ensures that the recorded brain activity is optimally suited for evaluating fatigue and cognitive workload.

Subsequently, the power spectrum of each epoch was computed using Welch’s method [50]. From this spectrum, the Alpha, Beta, and Theta band powers were extracted and averaged across the selected channels. These averages were used to compute the mental fatigue index (*MFI*), defined as the ratio of band powers [51], providing a quantitative measure of cognitive fatigue associated with the tasks:(1)$$MFI=\frac{\alpha +\theta }{\beta },$$
as well as the *Theta/Alpha ratio*(2)$$Theta/Alpha\;ratio=\frac{\theta }{\alpha }$$
the *Beta/Alpha ratio*(3)$$Beta/Alpha\;ratio=\frac{\beta }{\alpha }$$
and, lastly, the *Alpha ratio*(4)$$Alpha\;ratio=\frac{\alpha }{\alpha +\beta +\theta }$$
where α,β, and θ in Equations (Equation 1)–(Equation 4) refer to the powers calculated over the respective frequency bands. These indices were thus computed for each epoch so as to build up distributions with a congruous number of elements per subject.

The variables α, β, and θ in Equations (Equation 1)–(Equation 4) represent the powers measured across their respective frequency bands. For each epoch, these indices were computed to create distributions containing a sufficient number of data points per subject. This approach facilitated a comprehensive statistical analysis of the frequency band powers and their variations among subjects.

## 3. Results

### 3.1. Classification Results

The classification results for the ErrP and SSVEP paradigms are detailed in Table 2 and Table 3, respectively. Table 2 presents the validation accuracy values for the ErrP events, obtained using the Bayesian linear discriminant analysis (BLDA) classifier through a process of 5-fold cross-validation. Conversely, Table 3 displays the validation accuracy results for the SSVEP events, which were classified using the linear discriminant analysis (LDA) following a more rigorous 10-fold cross-validation.

The results summarized in Table 2 indicate the BLDA classifier’s validation accuracy for all subjects. Although it is relatively low compared with other BCI paradigms, it still aligns with the existing literature [52]. In contrast, the classification accuracy for SSVEP, as shown in Table 3, is significantly higher than that of ErrPs, highlighting a marked improvement in performance for this particular paradigm.

### 3.2. Task Achievement

The objective of measuring overall task achievement is to enable the robot to avoid obstacles that are not detected by its sensors. In the passive control scenario, the user passively observes the environment, while the system detects ErrPs. When a correctly classified ErrP is detected, the robot autonomously avoids the obstacle. In contrast, during active control, the user directly guides the robot by focusing on flickering stimuli such as SSVEPs to execute navigation commands, steering it away from obstacles. Each avoidance trial consists of three correct movements: left, right, and then left again (or the opposite sequence). Consequently, task completion is defined as a single event in the passive scenario and three sequential events in the active scenario, as illustrated in Figure 1. Table 4 presents the overall task achievement accuracy for both active and passive BCI control.

### 3.3. Mental Load Assessment

Figure 6, Figure 7, Figure 8, Figure 9, Figure 10, Figure 11, Figure 12, Figure 13, Figure 14 and Figure 15 show the obtained result from calculating mental load based on the *MFI*, *Alpha ratio*, *Beta/Alpha ratio*, the *Theta/Alpha ratio*.

## 4. Discussion

The results summarized in Table 2 show that while the validation accuracy for ErrPs is lower compared with other BCI paradigms, it remains consistent with findings in the existing literature [52]. When comparing classification performance for each event, SSVEPs demonstrate significantly higher accuracy than ErrPs. As shown in Table 3, SSVEP classification accuracy ranges from 68.7% (S9) to 84.9% (S1), whereas ErrP classification accuracy is considerably lower, varying between 46.0% (S9) and 72.9% (S2). Despite this difference, the passive ErrP-based approach results in higher overall task success in obstacle avoidance, highlighting its practical effectiveness.

This discrepancy arises due to the engagement required in each paradigm. SSVEP relies on the user actively attending to flickering stimuli, which enhances classification accuracy by maintaining focus. However, this sustained engagement demands continuous effort and precise execution of multiple commands. To successfully avoid an obstacle, for example, a subject must sequentially issue three correct commands—turn left, turn right, and move forward. Given the classification accuracy variability (e.g., S9 achieving 68.7% in SSVEP), errors in one or more steps can significantly reduce task success.

Conversely, the passive ErrP approach requires only a single correct classification to trigger an autonomous correction. This means that while SSVEP provides precise, high-speed control, its success rate is affected by the need for multiple accurate decisions. In contrast, the ErrP paradigm allows the system to handle avoidance autonomously with minimal user effort. This explains why, despite its lower classification accuracy, the passive method outperforms active control in overall task achievement, particularly for subjects who struggle to maintain high engagement over extended interactions. Moreover, the ErrP method requires a long training period, and the classifier does not perform well in generic classification as it is highly subject-specific. This means that the system needs to be tailored to each user, which can be time-consuming and less efficient. Enhancing the system to recognize varying levels of danger and improving the classifier’s generalization across different users would make the ErrP method more effective and user-friendly. Referring to the assessment of mental load, Figure 6 presents a comparative analysis of the MFI between ErrPs and SSVEPs. It reveals an overall increase in the MFI index for all participants from the start to the end of the experiment. Notably, exceptions were observed for subjects S7 in the ErrP condition and subjects S4, S5, S7, and S10 in the SSVEP condition. Generally, SSVEP registered lower MFI values compared with ErrP, likely due to the inclusion of all stimulus frequencies within the Beta band, which effectively reduces the MFI values.

Figure 7 illustrates the *Beta/Alpha* index for both ErrP and SSVEP. A general decrease in this index was observed for all subjects from the start to the end of the experiment, except for subjects S10 during ErrP and subjects S5 and S11 during SSVEP. The observed increment in the ErrP and decrement in the SSVEP can be attributed to the increasing Alpha power during fatigue, which indicates feelings of sleepiness or losing attention. Overall, SSVEP displayed higher *Beta/Alpha* values than ErrP because the Beta band contains all SSVEP stimulus frequencies.

Figure 8 compares the *Theta/Alpha* index for ErrPs and SSVEP. Most subjects experienced a decrease in this index during ErrP, except for subjects S4 and S10, while an increase was generally observed during SSVEP, excluding subjects S3 and S4.

Figure 9 shows the *Alpha* ratio, which generally increased from the start to the end of the experiment for most subjects. Exceptions include S10 during ErrP and subjects S4, S5, and S10 during SSVEP, who did not follow this trend.

Figure 10 and Figure 11 illustrate the mental fatigue index (MFI) against the *Beta/Alpha* index for ErrP and SSVEP data, respectively. In both cases, the MFI generally rises from the experiment’s start to end, except for subject S10 in ErrP and subjects S4, S5, S7, and S10 in SSVEP. The *Beta/Alpha* index typically decreases, with exceptions for subjects S4, S5, and S10 in SSVEP. This pattern reflects an increase in Alpha power, which drives up the MFI and lowers the *Beta/Alpha* index. Similar trends are evident in Figure 12 and Figure 13, which link MFI to the *Theta/Alpha* index for ErrP.

Figure 14 shows that most subjects experienced an increase in Alpha power by the experiment’s end, with the ErrP task consistently showing higher index values than the SSVEP task, indicating the former’s potentially more monotonous nature and its tendency to reduce engagement over time. Conversely, Figure 15 highlights that during the SSVEP task, the majority of subjects showed a greater increase in the *Theta/Alpha* index, suggesting that the active BCI task imposes a higher cognitive load and results in more pronounced shifts in brain activity related to mental workload or fatigue.

The indices used in this study, specifically *Theta/Alpha*, *Beta/Alpha*, and *(Theta + Alpha)/Beta*, are designed to estimate mental load or fatigue by analyzing power distribution across different EEG frequency bands. Particularly, the *(Theta + Alpha)/Beta* index offers a comprehensive view by considering the combined power of the Theta and Alpha bands relative to the Beta band. This approach captures a broader spectrum of neural activity, potentially providing a more nuanced assessment of mental workload or fatigue than the *Theta/Alpha* index alone.

Results from the study reveal notable differences in the MFI across most participants throughout the experiment. However, exceptions were observed with subjects S7 and S10, who displayed higher MFI values initially, which decreased by the experiment’s end. Typically, the overall MFI values for SSVEPs were lower than those for ErrPs among the majority of participants, attributable to the differences in the frequency bands used in the MFI calculations.

For SSVEPs, the Beta bandwidth, which includes all the stimulus frequencies (e.g., 12 Hz, 15 Hz, and 20 Hz), is part of the MFI equation’s numerator. This inclusion dilutes the power ratio, consequently lowering the MFI value for SSVEP. Conversely, in the case of ErrPs, the Theta bandwidth is included in the MFI equation’s denominator, which tends to elevate the MFI index value for ErrPs, reflecting higher mental fatigue levels [53].

It is important to consider that the observed differences in the index values between SSVEPs and ErrPs could be shaped by multiple factors, including the specific experimental design, the placement of the electrodes, and individual variations in EEG responses. A thorough understanding of these factors is crucial for correctly interpreting the results and considering the implications when comparing different BCI paradigms or cognitive tasks.

## 5. Conclusions

This study identifies the most effective paradigm to be applied to mobile robot navigation by integrating human feedback via a brain–computer interface to compensate robot errors in terms of undetected obstacles. The BCI system operates in both passive and active modes: the passive BCI relies on error-related potentials, while the active BCI utilizes steady-state visually evoked potentials, allowing users to interact with the system through both neural and traditional input mechanisms.

The proposed protocol supports real-time feedback by incorporating a human observer into the control loop. The mental fatigue index, designed to stimulate ErrPs, effectively corrects navigation errors with a lower cognitive effort than SSVEPs, thereby reducing mental fatigue. However, prolonged passive engagement can lead to decreased alertness, as evidenced by increased Alpha power [54]. Conversely, the active SSVEP method, while more mentally demanding, maintains higher engagement over time, as indicated by increased Theta and Alpha band activity. These effects are particularly evident in Figure 9 and Figure 10, which illustrate distinct spectral changes for ErrP and SSVEP conditions [53].

A key finding is that engagement significantly impacts classification accuracy. While SSVEP provides precise, real-time control, it requires sustained attention and multiple sequential commands for effective obstacle avoidance. For example, avoiding an obstacle may require three consecutive commands: turn left, turn right, and move forward. Any misclassification within this sequence reduces overall success. In contrast, the passive ErrP method requires only a single correct classification, enabling the robot to autonomously adjust its path. Despite its lower classification accuracy, the passive approach ultimately achieves higher task success as it minimizes cognitive load and reliance on user engagement.

Our results align with prior research, indicating increased Alpha power under higher cognitive demands and increased frontal Theta in response to workload [54]. Given the limitations of EEG spectral analysis in fatigue detection, future studies should explore entropy-based methods, which have shown greater consistency in assessing cognitive fatigue [55].

A major limitation of this study is the lack of generalization across subjects due to insufficient data. The current dataset does not provide enough diversity to train sophisticated deep learning models such as recurrent neural networks (RNNs) or convolutional neural networks (CNNs). Consequently, the current classifier remains highly subject-dependent, requiring extensive individual calibration, which increases training and setup time in real-world applications.

To address this limitation, future work should focus on collecting data from a larger pool of subjects with significantly more trials per participant. A broader dataset would allow for training models that generalize across individuals rather than being fine-tuned to a single user. Additionally, generative models and data augmentation techniques could be leveraged to expand the dataset artificially, improving model robustness and reducing overfitting. With a sufficiently large dataset, deep learning methods could be applied to develop a generic classifier that adapts to different users without requiring extensive retraining. This would drastically reduce system calibration time and enhance usability in practical scenarios.

## Figures and Tables

**Figure 1 brainsci-15-00359-f001:**
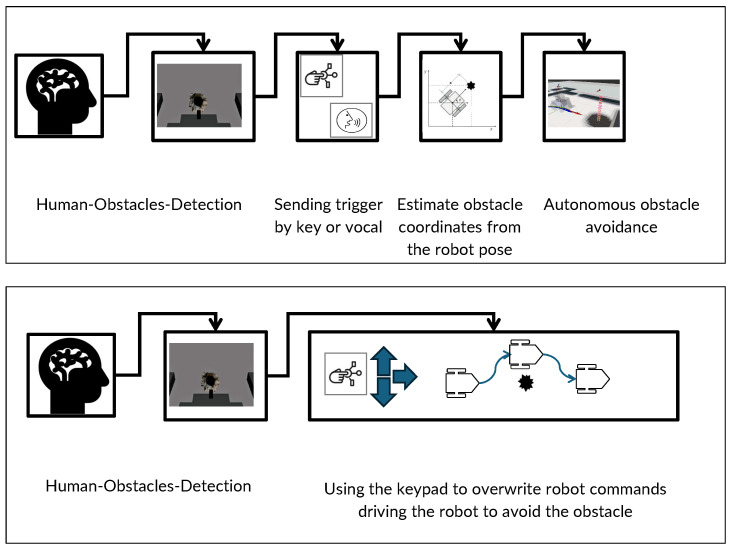
Illustration of control mechanisms in robotic navigation without BCI: the upper panel shows *indirect control*, where the human operator visually detects obstacles and sends a trigger to the robot’s system for autonomous avoidance. The lower panel illustrates *direct control*, where the user manually navigates the robot using a keyboard to avoid detected obstacles.

**Figure 2 brainsci-15-00359-f002:**
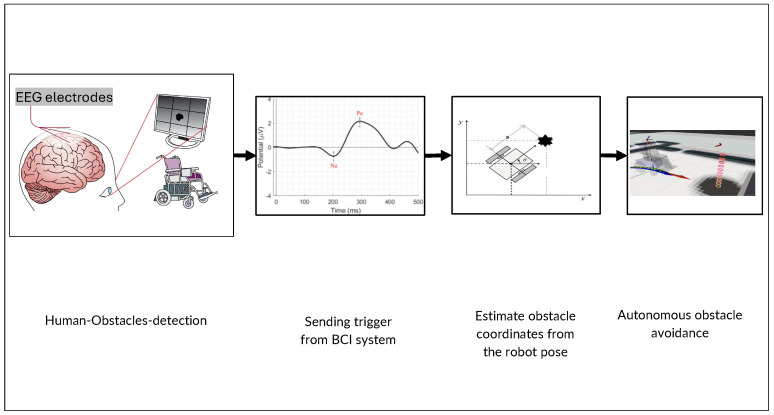
Proposed “Passive BCI”, modified from the original *indirect robot control*, integrating ErrPs to detect user-perceived errors during robot navigation.

**Figure 3 brainsci-15-00359-f003:**
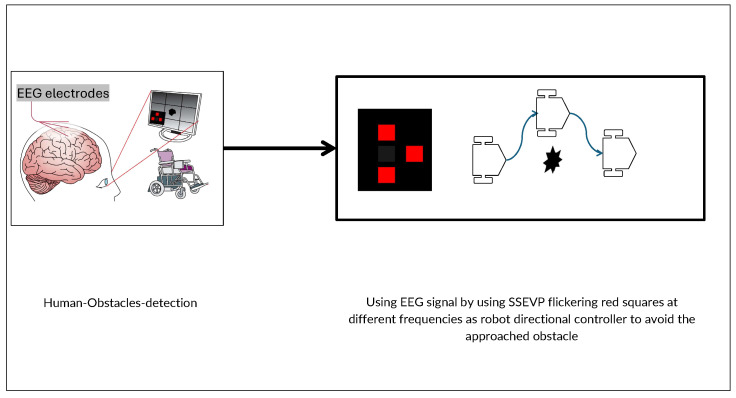
Proposed “Active BCI”, modified from the original *direct robot control*, incorporating SSVEPs to enable active user control during robot navigation.

**Figure 4 brainsci-15-00359-f004:**
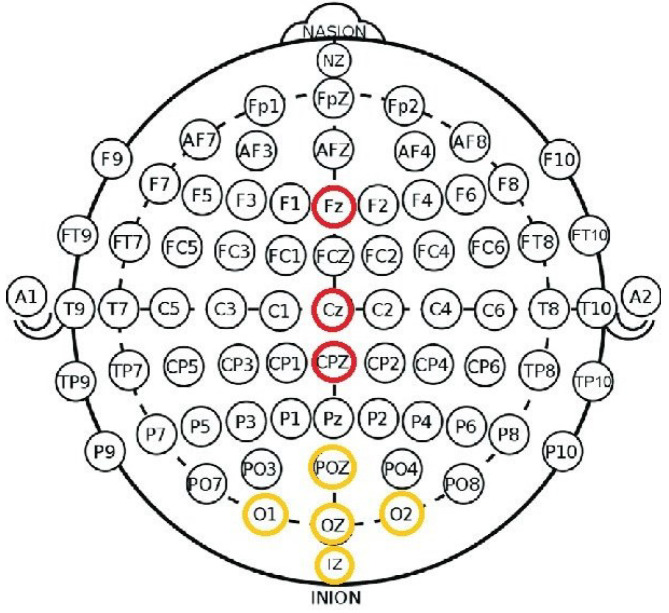
EEG channels configuration: 8 channels were placed on the scalp according to the 10/20 standard system for EEG recording.Channels with red circle was dedicated for ErrPs, and the yellow is used for SSVEP.

**Figure 5 brainsci-15-00359-f005:**
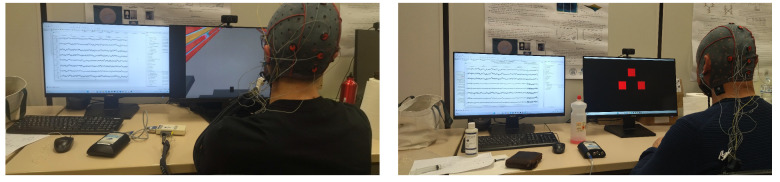
(**Left**) data acquisition for the passive BCI during the ErrP task; (**Right**) data acquisition for the active BCI during the SSVEP task.

**Figure 6 brainsci-15-00359-f006:**
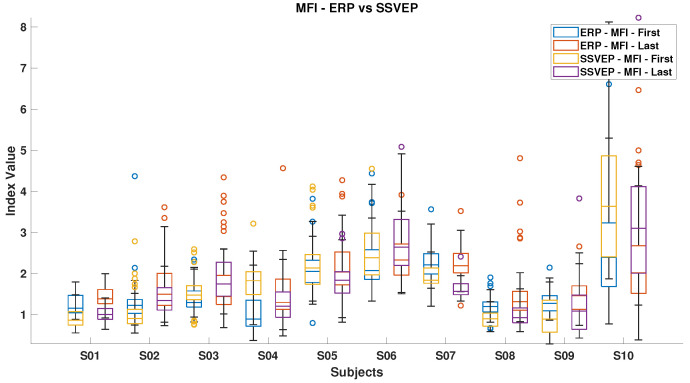
Mental fatigue index (*MFI*) in ErrPs and SSVEPs, highlighting the corresponding fatigue levels for each event type.

**Figure 7 brainsci-15-00359-f007:**
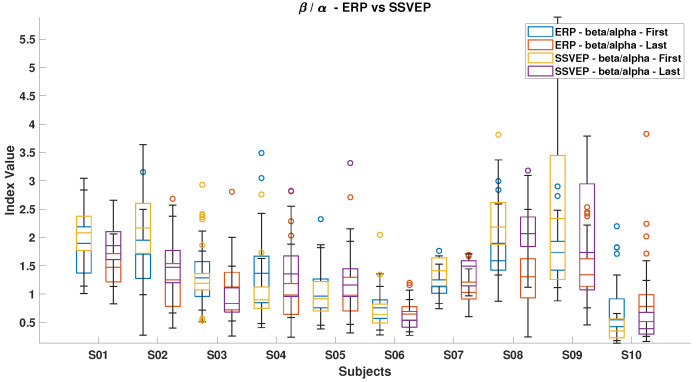
*Beta/Alpha ratio* in ErrPs and SSVEPs, highlighting the differences in brainwave patterns between these two types of responses.

**Figure 8 brainsci-15-00359-f008:**
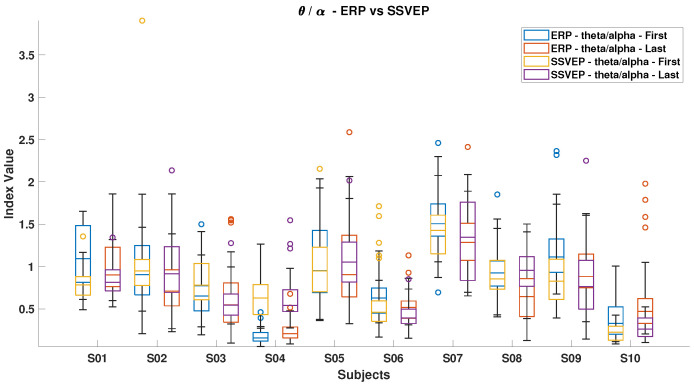
*Theta/Alpha ratio* in ErrPs and SSVEPs, comparing the relative brainwave activities in these scenarios.

**Figure 9 brainsci-15-00359-f009:**
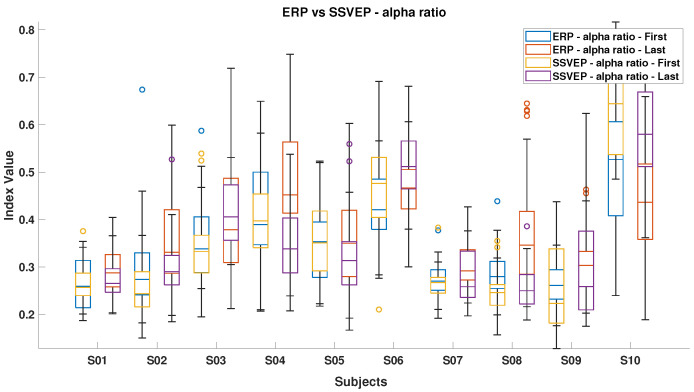
*Alpha ratio* index in ErrPs and SSVEPs, showcasing variations in brainwave activity between these two conditions.

**Figure 10 brainsci-15-00359-f010:**
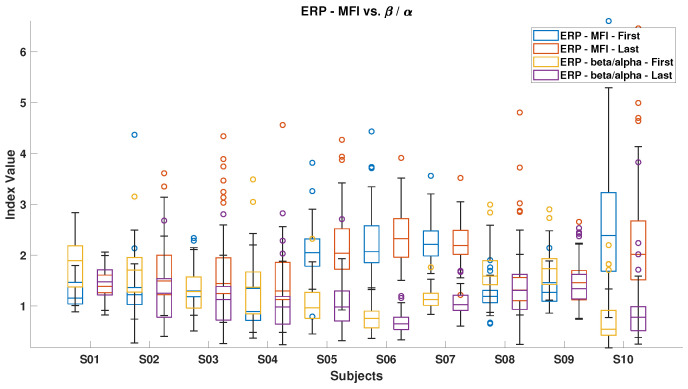
*MFI* and *Beta/Alpha ratio* for ErrPs, which showcases the mental fatigue index and *Beta/Alpha ratio* specifically for error-related potentials (ErrPs).

**Figure 11 brainsci-15-00359-f011:**
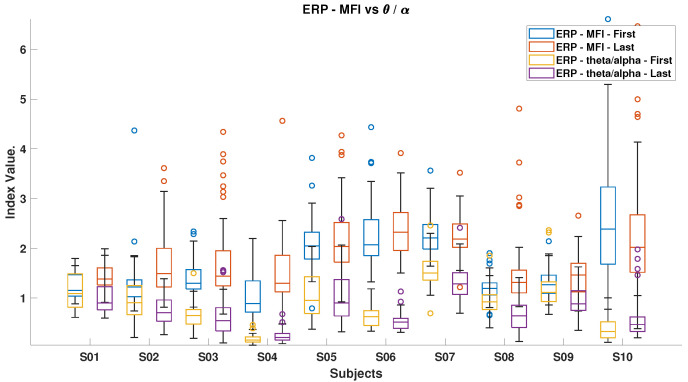
*MFI* and *Theta/Alpha ratio* for both ErrPs and SSVEPs.

**Figure 12 brainsci-15-00359-f012:**
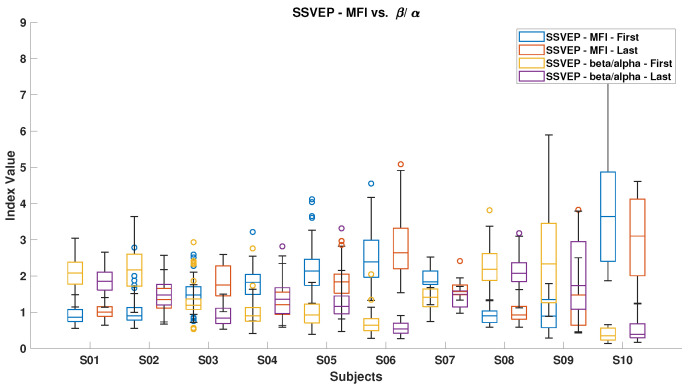
*MFI* and *Beta/Alpha ratio* for SSVEP, which depicts the *MFI* and *Beta/Alpha ratio*, specifically for SSVEP.

**Figure 13 brainsci-15-00359-f013:**
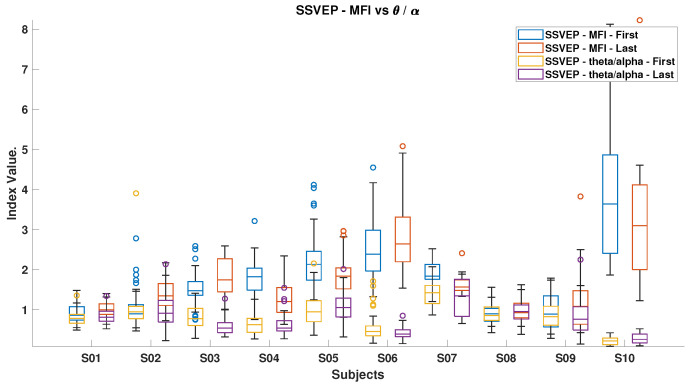
*MFI* and *Theta/Alpha* Indices for SSVEPs, which illustrate the m*MFI* and *Theta/Alpha ratio* in the context of SSVEPs.

**Figure 14 brainsci-15-00359-f014:**
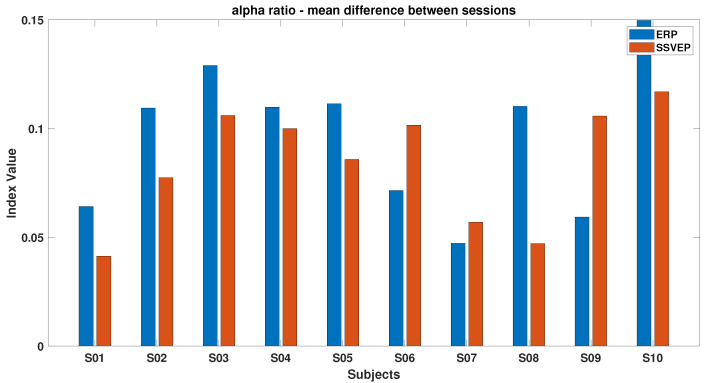
The mean of the differences between beginning and ending values of the *Alpha ratio* index for ErrPs and SSVEPs, containing alpha brainwave power to other brainwave frequencies during error-related potentials (ErrPs) and SSVEPs.

**Figure 15 brainsci-15-00359-f015:**
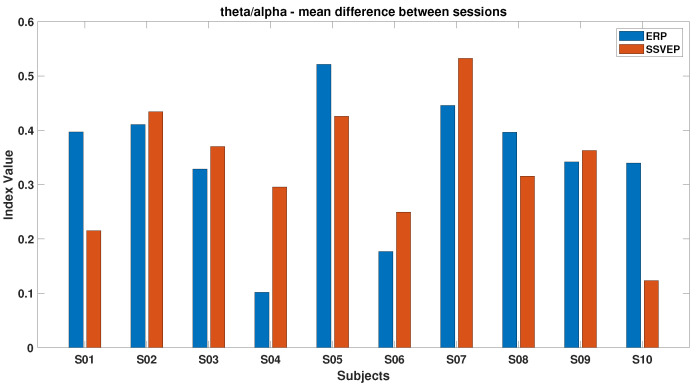
The mean of the differences between beginning and ending values of the *Theta/Alpha ratio* for ErrPs and SSVEPs; ratio between Theta and Alpha brainwave frequencies in the context of both ErrPs and SSVEPs.

**Table 1 brainsci-15-00359-t001:** Summary of EEG frequency bands’ properties, including frequency range, amplitude, typical brain location, and associated mental states.

Band	Frequency (Hz)	Amplitude (µV)	Location	Activity
Delta	0.5–4	100–200	Frontal	Deep sleep
Theta	4–8	5–10	Various	Drowsiness, light sleep
Alpha	8–13	20–80	Posterior region of head	Relaxed
Beta	13–30	1–5	Symmetrical distribution, most evident frontally	Active thinking, alert

**Table 2 brainsci-15-00359-t002:** ErrP classification accuracy from 10 healthy subjects.

Subject	Avg Classification Accuracy %
S1	57.6
S2	72.9
S3	51.6
S4	52.0
S5	53.8
S6	59.6
S7	58.5
S8	51.0
S9	46.0
S10	53.8

**Table 3 brainsci-15-00359-t003:** SSVEP classification accuracy from 10 healthy subjects.

Subject	Avg Classification Accuracy %
S1	84.9
S2	75.7
S3	77.6
S4	76.2
S5	77.5
S6	75.1
S7	78.9
S8	78.2
S9	68.7
S10	75.3

**Table 4 brainsci-15-00359-t004:** Task achievement accuracy for active and passive methods.

Subject	Active Task Achievement Accuracy	Passive Task Achievement Accuracy
S1	61	57
S2	43	71
S3	47	51
S4	44	51
S5	47	53
S6	42	59
S7	49	58
S8	48	50
S9	32	45
S10	43	53

## Data Availability

The dataset used for this study can be obtained from the corresponding author upon reasonable request (the data is currently being used for ongoing research and is undergoing formatting to align with standard EEG dataset conventions).

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
