# Peer review of "Real-Time Mobile Robot Obstacles Detection and Avoidance Through EEG Signals"

_brainsci, 2025, doi:10.3390/brainsci15040359_

Round 1

Reviewer 1 Report

Comments and Suggestions for Authors

This paper investigated current control paradigms of BCI for mobile robot obstacle detection with ErrP for a passive approach and SSVEP for an active approach. The findings showed passive BCIs cause lower mental fatigue and lower engagement levels. I believe the findings are valuable, but the manuscript needs improvement.

Line 16: ‘(SSVEP)..’

Line 19: ‘of85%’

I think the introduction is superficial. It does not provide any detailed discussion of relevant previous studies. It only briefly mentions the significance of BCIs and their paradigms. The authors should introduce specific prior work or studies that have directly addressed related challenges in human-robot interaction, obstacle detection, or navigation using BCIs.

Technical details are missing.

While the authors provided a brief rationale for the chosen EEG electrodes and I also believe these are generally considered regions, I think it is not sufficient. Please provide evidence for the electrodes and explain how the regions is associated with the cognitive process and tasks involved in the study.

Also, PO7?

Please provide the spec of the filter used like filter order, type, window type, …

ICA is a linear decomposition. It does not itself identify and eliminate muscle and eye artifacts. You should provide enough details and standards.

The authors did not mention data rank. According to the official EEGLAB documentation, the pca option for ‘runica’ can be used to match the rank and ensure full-ranked data. I believe it is likely that you used this, and then it would be good to explicitly mention this like “we used the pca option in ‘runica’ to ensure full-ranked data [a relevant paper]”. I think you can find a suitable paper about the details.

https://sccn.ucsd.edu/wiki/Makoto's_preprocessing_pipeline#A_study_on_the_ghost_IC_was_published_.28added_on_04.2F04.2F2023.29

The results are not validated with appropriate statistical tests. I highly recommend putting them. Otherwise, it would be difficult to consider valid.

The discussion is too focused on metrics like accuracy. While they are of course necessary, the discussion should consider real-world applications and situations.

In addition, the discussion should address broader contexts. For instance, a robot's movements can influence human cognition in various ways, depending on specific aspects of the movement involved, as demonstrated in [1]. It would be valuable to consider how specific movement characteristics like precision and speed might impact cognitive engagement, providing insights that could guide future studies.

[1] https://doi.org/10.3390/s23010277

Author Response

Comments 1:[  Line 16: ‘(SSVEP)..’  Line 19: ‘of85%’]

Response 1: [Thank you for your valuable feedback. We appreciate your recognition of the study’s contributions. In response to your comments, we have thoroughly revised and improved the introduction, discussion, and conclusion sections to enhance clarity and provide a more comprehensive analysis of our findings.

Additionally, we have corrected the typographical errors in Line 16 (‘(SSVEP)..’) and Line 19 (‘of85%’) as noted.

We appreciate your constructive feedback and believe these revisions strengthen the manuscript.]

Comments 2: [I think the introduction is superficial. It does not provide any detailed discussion of relevant previous studies. It only briefly mentions the significance of BCIs and their paradigms. The authors should introduce specific prior work or studies that have directly addressed related challenges in human-robot interaction, obstacle detection, or navigation using BCIs.]

 Response 2:[ Thank you for your feedback. We acknowledge that the introduction could be expanded to provide a more detailed discussion of relevant prior studies. In our revised manuscript, we have enhanced the introduction by including a more in-depth review of key BCI paradigms, which can generally be categorized as active, passive, and.

Additionally, we have incorporated references to previous studies that specifically address human-robot interaction, obstacle detection, and navigation using BCIs, highlighting their challenges and contributions. This ensures a clearer contextual foundation for our work. We appreciate your suggestion and believe that these revisions will provide a stronger background for our study.]

Comments 3:[ Technical details are missing. While the authors provided a brief rationale for the chosen EEG electrodes and I also believe these are generally considered regions, I think it is not sufficient. Please provide evidence for the electrodes and explain how the regions are associated with the cognitive process and tasks involved in the study.

Also, PO7?]

Response 3:[ Thank you for your feedback. We selected the EEG electrode configuration (Fz, Cz, CPz, Oz, PO7, O1, O2) based on prior research recommendations for capturing both error-related potentials (ERPs) in the frontal cortex and steady-state visual evoked potentials (SSVEP) in the occipital region.

For error-related potentials (ERPs), frontal electrodes (e.g., Fz, Cz, CPz) are well-established in detecting error-related negativity (ERN) and error positivity (Pe), which are linked to cognitive control and performance monitoring (Gehring et al., 1993; Falkenstein et al., 2000).

For SSVEP, occipital electrodes (e.g., Oz, O1, O2, PO7) are commonly used due to their sensitivity to visual stimulus-driven neural oscillations (Norcia et al., 2015). PO7, in particular, has been used in studies focusing on lateralized visual processing and SSVEP-based brain-computer interfaces (BCI) (Zhang et al., 2010).

We will include these references in the manuscript to strengthen our justification.]

Comments 4:[ Please provide the spec of the filter used like filter order, type, and window type, …]

Response 4:[ Thank you for your comment. In  the revised  version of the manuscript, we provide more details about the filtering process in the preprocessing section. For ErrP event classification, we applied a two-phase temporal filtering approach:

"Initially, a second-order Butterworth low-pass filter was applied, followed by decimation, reducing the original 256 Hz sampling rate to 32 Hz. Subsequently, a second-order high-pass Butterworth filter was applied. The resulting signal, with a bandwidth between 1 to 10 Hz, was optimal for capturing ErrP activities. EEG data were then epoched in time windows from -1 s to 1 s, with zero marking the key press."

For SSVEP classification, we used the Common Spatial Pattern (CSP) algorithm to extract features from filtered signals:

"Butterworth filters were applied to each stimulus frequency, with low and high cut-off values set 0.25 Hz below and above the respective stimulus frequency. The CSP algorithm transformed multi-channel data into a two-channel format, optimizing class separation by maximizing variance in one class while minimizing it in another. The extracted feature matrices were then classified using Linear Discriminant Analysis (LDA). Each trial  epoched  is  5 seconds,  with a moving  window length of  0.5-second for classification." We will ensure that these details are presented in the preprocessing section of the manuscript.]

Comments 5:[ ICA is a linear decomposition. It does not itself identify and eliminate muscle and eye artifacts. You should provide enough details and standards. The authors did not mention data rank. According to the official EEGLAB documentation, the PCA option for ‘runica’ can be used to match the rank and ensure full-ranked data. I believe it is likely that you used this, and then it would be good to explicitly mention this like “we used the PCA option in ‘runica’ to ensure full-ranked data [a relevant paper]”. I think you can find a suitable paper about the details.

https://sccn.ucsd.edu/wiki/Makoto's_preprocessing_pipeline#A_study_on_the_ghost_IC_was_published_.28added_on_04.2F04.2F2023.29 ]

Response 5[:  Thank you for your insightful comment. You are correct that ensuring full-ranked data is important when using ICA. In our manuscript, we explicitly mention that we used the pca option in runica to match the data rank and ensure full-ranked data as it has been recommended to compute average reference after ICA. We have also included a relevant citation to support this choice.]

Comments 6:[ The results are not validated with appropriate statistical tests. I highly recommend putting them. Otherwise, it would be difficult to consider valid.]

Response 6:[  Thank you for your valuable feedback. We fully acknowledge the importance of statistical validation in ensuring the reliability and robustness of results. In this study, however, the primary objective is to obtain preliminary evidence on which paradigm proves most effective by considering factors such as mental load, task achievement accuracy, and event classification accuracy.

Rather than focusing on a direct comparison between specific methods or classifiers, we aim to explore the broader feasibility of different paradigms and their impact on user performance and cognitive effort. While we have not conducted extensive statistical tests at this stage, these preliminary findings serve as a foundation for future studies, where more rigorous statistical analyses will be applied to further validate and refine our results.

We greatly appreciate your suggestion, and we will ensure that statistical validation becomes a key component in our upcoming investigations.]

Comment 7:[ The discussion is too focused on metrics like accuracy. While they are of course necessary, the discussion should consider real-world applications and situations.

In addition, the discussion should address broader contexts. For instance, a robot's movements can influence human cognition in various ways, depending on specific aspects of the movement involved, as demonstrated in [1]. It would be valuable to consider how specific movement characteristics like precision and speed might impact cognitive engagement, providing insights that could guide future studies.

[1] https://doi.org/10.3390/s23010277  ]

Response 7:[  We sincerely thank the reviewer for their insightful comment. You are correct that a robot's movement characteristics such as speed and precision can influence human cognition and error perception.

Our current study investigates how the human brain responds to different classes of robot errors, ranging from minor to severe while considering variations in the robot’s speed and distance. We aim to explore whether error-related potentials (ErrPs) show different amplitudes as the perceived danger increases or if these signals are modulated by movement dynamics.

However, the present BCI paradigm is designed for binary error detection distinguishing whether a navigation error has occurred making it challenging to directly assess how movement patterns impact error perception within this setup. Nonetheless, we are preparing data for future work, where a key objective will be to examine how ErrPs vary across different robot errors, further addressing the relationship between movement dynamics and cognitive responses.

We greatly appreciate the reviewer’s recommendation, as it provides valuable direction for our ongoing and future research.]

Reviewer 2 Report

Comments and Suggestions for Authors

1. The objectives of the study are very relevant and the approaches for solving these problems are quite adequate

2. the classification accuracy based on the results of EEG analysis depends on many factors: 1) alpha peak frequency ( as an indicator of neuronal efficiency and sensorimotor integration), 2) hormonal state ( women in low-hormonal phases have lower EEG frequency and consequently lower cognitive efficiency than in high-hormonal phases), 3) psycho-emotional stress, a marker of which is the scalp EMG, and which through affecting both the source of electrical signal generation and contaminating the EEG, can also reduce the accuracy of prediction. Therefore, the variability of the results obtained may depend on these factors. 

3- The authors should state in the methods section what frequency pattern at baseline of the subjects, how many women and what hormonal states (menstrual ctcle phase) were they on? 

4. Minor comments: 

a. what First-Person View means 

b. the abbreviation ( ErrPs) in abstracts should be deciphered first before using in text 

Author Response

Comments 1: [ Factors: 1) alpha peak frequency ( as an indicator of neuronal efficiency and sensorimotor integration), 2) hormonal state ( women in low-hormonal phases have lower EEG frequency and consequently lower cognitive efficiency than in high-hormonal phases), 3) psycho-emotional stress, a marker of which is the scalp EMG, and which through affecting both the source of electrical signal generation and contaminating the EEG, can also reduce the accuracy of prediction. Therefore, the variability of the results obtained may depend on these factors.]

Response 1: [Thank you for your insightful comment. We acknowledge that several factors, including alpha peak frequency, hormonal state, and psycho-emotional stress, can influence EEG-based classification accuracy. Specifically, hormonal fluctuations—such as the lower EEG frequency observed in women during low-hormonal phases—may impact cognitive efficiency, while psycho-emotional stress, often reflected in scalp EMG activity, can further contaminate EEG signals and affect prediction accuracy. To mitigate these effects, our preprocessing pipeline includes targeted bandpass filtering. Specifically, for ErrPs, we applied a passband of 3.5–8.5 Hz, while for SSVEP, a bank filter was centered around each flicker frequency (12 Hz, 15 Hz, and 20 Hz). Additionally, for mental load analysis, we utilized frequency-based metrics that inherently minimize EMG interference by focusing on specific EEG bands: theta (4–7 Hz), alpha (8–12 Hz), and lower beta (13–20 Hz). Given that EMG artifacts typically fall within the 10–500 Hz range, our filtering approach effectively reduces their impact, ensuring cleaner EEG signals and more reliable results. Furthermore, to account for baseline variability, we measured two minutes of resting-state EEG activity at the beginning of each session, where participants remained still and did not engage in any task. This baseline measurement was used for both ErrPs and SSVEP conditions to establish a reference point before task-related activity. We recognize the importance of factors such as hormonal state and psycho-emotional stress in EEG variability and plan to further investigate their influence in future studies.]

Comments 2: [The authors should state in the methods section what frequency pattern at baseline of the subjects, how many women and what hormonal states (menstrual cycle phase) were they on?]

Response 2:[ Thank you for your insightful comment. In our current study, we included 10 subjects, of whom 4 were women. However, we did not account for the hormonal state (e.g., menstrual cycle phase) in this dataset. We acknowledge that factors such as hormonal fluctuations and subject mood may influence EEG signals, making this an interesting aspect to explore. As we expand our study to 30 subjects, it could be possible to incorporate a questionnaire to collect relevant information on hormonal states and mood, allowing for a more comprehensive analysis in future work. We appreciate your valuable suggestion and will consider it in our ongoing data collection.]

Comments 3: [Minor comments:

  1. what First-Person View means
  2. the abbreviation ( ErrPs) in abstracts should be deciphered first before using in text ]

Response 3:[ Thank you for your careful review and helpful suggestions. We have addressed both comments as follows:

  1. We have clarified the meaning of First-Person View (FPV) in the manuscript, explaining that it refers to a perspective where visual information is presented from the viewpoint of the user, often used in virtual reality (VR), robotics, and drone applications.
  2. We have ensured that the abbreviation ErrPs (Error-Related Potentials) is introduced and defined before its first use in the abstract.

We appreciate your feedback and believe these corrections enhance the clarity of our manuscript.]

Reviewer 3 Report

Comments and Suggestions for Authors

The article explores the use of EEG-based BCIs for real-time obstacle detection and avoidance in mobile robots, particularly focusing on assistive technologies like wheelchairs. It investigates two paradigms: passive BCIs utilizing ErrPs for automated error correction and active BCIs employing SSVEPs for direct user control. The study evaluates these methods based on classification accuracy, user mental workload, and engagement levels. While passive BCIs demonstrated lower cognitive load and easier implementation, active BCIs achieved higher accuracy and engagement, though at the cost of greater mental fatigue. The findings highlight critical trade-offs between system complexity, user cognitive effort, and interaction quality, offering insights for improving the design of assistive robotic systems that prioritize both functionality and user experience. This study is interesting and has its own significance; however, I have the following major concerns regarding some aspects of this article:

1.     The article lacks a detailed comparison of the proposed method with other state-of-the-art BCI approaches used in robotic navigation. Including such comparisons could provide stronger evidence of the novelty and effectiveness of the proposed system.

2.     The study does not address inter-subject variability in EEG signals, which is critical for generalizing the findings. A detailed discussion on how the system could handle such variability (e.g., through transfer learning or adaptive algorithms) is necessary.

3.     The methodology for mental workload assessment relies heavily on spectral analysis indices, which may not capture fatigue comprehensively. Incorporating entropy-based metrics or real-time adaptive workload evaluation could strengthen the findings.

4.     The classification accuracy of ErrPs (max 73%) is significantly lower compared to SSVEPs (up to 85%), which may limit its practical usability. Further optimization of the ErrP classifier and inclusion of advanced machine learning techniques should be considered.

5.     The study employs a limited number of EEG channels (8 channels), which may not capture sufficient spatial resolution for reliable classification. Expanding the channel set or justifying the choice with additional analysis could strengthen the methodology.

6.     There is no discussion on signal artifacts, such as those caused by eye movements or muscle activity, and how these were mitigated in preprocessing. A clear description of artifact removal methods and their effectiveness would improve the robustness of the presented methodology.

7.     The classification models used (BLDA and LDA) are relatively simple compared to modern deep learning approaches. Exploring advanced classifiers, such as convolutional neural networks (CNNs) or recurrent neural networks (RNNs), could improve accuracy and generalization.

8.     The choice of accuracy as the sole performance metric may not fully represent system effectiveness. Including additional metrics, such as precision, recall, and F1-score, would provide a more comprehensive evaluation of classification performance.

9.     The mental workload analysis lacks a connection to task performance metrics. Correlating workload indices with navigation success rates or error frequencies would offer deeper insights into the usability of the system.

Author Response

Comments 1: [The article lacks a detailed comparison of the proposed method with other state-of-the-art BCI approaches used in robotic navigation. Including such comparisons could provide stronger evidence of the novelty and effectiveness of the proposed system.]

Response 1: [Thank you for your valuable comment. In the revised version of the manuscript, we have included a more detailed discussion on state-of-the-art BCI approaches commonly used for robotic navigation and robot error correction. This addition provides a broader context for our work and highlights the novelty and effectiveness of our proposed system in comparison to existing methods.]

Comments 2:[ The study does not address inter-subject variability in EEG signals, which is critical for generalizing the findings. A detailed discussion on how the system could handle such variability (e.g., through transfer learning or adaptive algorithms) is necessary.]

Response 2:[ Thank you for your valuable comment. We acknowledge that inter-subject variability in EEG signals is a crucial factor for generalizing BCI systems. While recent research is exploring the feasibility of generic classifiers for BCI, achieving this requires large-scale datasets, both at the individual level and across a diverse subject pool.

In our current study, we focus on evaluating the applicability of the paradigms within subjects, as EEG signals can vary even within the same individual due to different moods, cognitive states, or external conditions. However, we recognize the importance of extending this investigation to a broader subject range. Moving forward, we plan to increase the number of subjects and trials to work toward more generalizable classifiers and paradigms. We appreciate your insightful suggestion and will consider it in our future research.]

Comments 3:[ The methodology for mental workload assessment relies heavily on spectral analysis indices, which may not capture fatigue comprehensively. Incorporating entropy-based metrics or real-time adaptive workload evaluation could strengthen the findings.]

Response 3: [Thank you for your insightful comment. We acknowledge that while spectral analysis indices provide valuable insights into mental workload, they may not fully capture fatigue dynamics. Based on our findings, we have concluded that future studies should incorporate entropy-based methods to enhance accuracy and differentiation. However, for this study, our primary objective was to demonstrate that passive and active paradigms can be effectively evaluated by weighting their mental load demands. The current approach provides a sufficient basis for comparing these methods, offering initial insights into their applicability. In future refinement studies, we plan to integrate entropy-based methods to better distinguish between workload-related activation, stimulus-induced responses (e.g., SSVEP, ErrPs), and fatigue-related changes, while also minimizing potential bandwidth interference. We appreciate your valuable suggestion and will take it into account in our future research.]

Comments 4:[ The classification accuracy of ErrPs (max 73%) is significantly lower compared to SSVEPs (up to 85%), which may limit its practical usability. Further optimization of the ErrP classifier and inclusion of advanced machine learning techniques should be considered.]

Response 4:[ Thank you for your insightful comment. The difference in classification accuracy between ErrPs (max 73%) and SSVEPs (up to 85%) is expected and can be justified by the fundamental nature of these signals. ErrPs are event-related potentials that arise only when a person perceives an error, making them inherently weaker and more variable due to their dependency on subjective perception and cognitive processing. In contrast, SSVEPs are generated in response to a steady visual stimulus with a distinct frequency, resulting in a more consistent and easily detectable neural response. With appropriate signal processing, SSVEP features can be reliably extracted and classified with high accuracy.

Despite the lower accuracy of ErrPs, their potential for hands-free, intuitive error correction in robotic systems remains highly promising. If further optimized, ErrP-based correction mechanisms could significantly enhance human-robot interaction by enabling seamless, effortless robot adjustments based on the user’s cognitive feedback. In future work, we plan to explore advanced machine-learning techniques to improve ErrP detection and classification, thereby increasing its practical usability.]

Comments 5:[ The study employs a limited number of EEG channels (8 channels), which may not capture sufficient spatial resolution for reliable classification. Expanding the channel set or justifying the choice with additional analysis could strengthen the methodology.]

Response 5: Thank you for your insightful comment. We have updated the article to explicitly mention the reasons behind our channel selection. As stated, while increasing the number of EEG channels can improve spatial resolution, our approach prioritizes developing a practical and efficient BCI system for real-world applications. Reducing the number of channels enhances system usability by lowering cost, reducing computational complexity, and improving ease of use in real-life scenarios. That said, we recognize the importance of capturing a more comprehensive neural response for optimization. Therefore, in our updated acquisition setup, we have expanded to 32 channels to improve spatial resolution. Our long-term goal remains to refine channel selection, identifying a minimal yet effective subset that maintains high classification performance. Ideally, we aim to achieve robust results with as few as one channel for passive paradigms and one channel for active paradigms, ensuring both efficiency and practicality. We appreciate your valuable suggestion and have revised the article accordingly to clarify our approach. We will continue working toward a balanced solution that optimizes both accuracy and real-world applicability in future research.]

Comments 6:[ There is no discussion on signal artifacts, such as those caused by eye movements or muscle activity, and how these were mitigated in preprocessing. A clear description of artifact removal methods and their effectiveness would improve the robustness of the presented methodology.]

Response 6: [Thank you for your valuable comment. In the revised manuscript, we have highlighted and elaborated on the artifact removal methods used in signal preprocessing. Specifically, we mitigate noise and artifacts—such as those caused by eye movements and muscle activity—by applying a targeted bandpass filter. This filter is carefully restricted to a minimal bandwidth that is sufficient to capture both ErrPs and SSVEP signals while preserving relevant frequency components, including alpha, delta, theta, and lower beta. This preprocessing approach effectively reduces unwanted high-frequency muscle artifacts (EMG) and low-frequency drifts, ensuring cleaner EEG signals without compromising the essential neural features. We appreciate your suggestion, as it strengthens the methodological clarity of the study.]

Comments 7:[ The classification models used (BLDA and LDA) are relatively simple compared to modern deep learning approaches. Exploring advanced classifiers, such as convolutional neural networks (CNNs) or recurrent neural networks (RNNs), could improve accuracy and generalization.]

Response 7: [Thank you for your insightful comment. In this preliminary study, our primary objective was to highlight the key differences between passive and active paradigms within a small group of subjects and a limited number of trials. Given these constraints, we opted for simpler classifiers (BLDA and LDA), which are well-established in EEG research and have been shown to perform reliably with smaller datasets. While deep learning approaches such as CNNs and RNNs offer potential advantages, they typically require large-scale datasets to effectively learn EEG patterns. Applying deep learning to small datasets can lead to overfitting and poor generalization. However, as we expand our study to 30+ subjects with a significantly larger number of trials, we plan to explore RNN-based models to improve classification accuracy and generalizability. We appreciate your valuable suggestion and will consider advanced classifiers in our future research as our dataset grows.]

Comments 8:[ The choice of accuracy as the sole performance metric may not fully represent system effectiveness. Including additional metrics, such as precision, recall, and F1-score, would provide a more comprehensive evaluation of classification performance.]

Response 8: [Thank you for your valuable suggestion. In this study, our primary focus was on evaluating the ability of a simple classifier to accurately detect events, making classification accuracy a sufficient primary metric for assessing overall system performance. However, we acknowledge the importance of additional evaluation metrics such as precision, recall, and F1-score, which provide deeper insights into classifier behavior, especially in cases of class imbalance.

In future work, as we expand our dataset and explore more advanced classification models, we plan to incorporate these additional metrics to offer a more comprehensive performance evaluation.

We appreciate your insightful feedback and will consider it in our ongoing research.]

Comments 9:[ The mental workload analysis lacks a connection to task performance metrics. Correlating workload indices with navigation success rates or error frequencies would offer deeper insights into the usability of the system.]

Response 9: [Thank you for your insightful comment. We acknowledge the importance of correlating mental workload indices with task performance metrics to provide a deeper understanding of the system's usability. Currently, we can measure task completion time for each subject and trial. Moving forward, we plan to enhance our analysis by incorporating overall task accuracy, which will involve integrating task execution performance during both passive control (robot accuracy and ErrP event trigger accuracy) and active control (where each task consists of three consecutive movements to navigate obstacles). This approach will allow for a more comprehensive evaluation of how workload levels impact navigation success rates and error frequencies, ultimately strengthening the system's usability assessment. We appreciate your valuable suggestion and will incorporate these metrics in future studies.]

Round 2

Reviewer 1 Report

Comments and Suggestions for Authors

The issues have been addressed.

Reviewer 3 Report

Comments and Suggestions for Authors

The authors have satisfactorily addressed my concerns. I have no further comments to raise and recommend this article for acceptance. Best of luck to the authors.